# Interleukin-4 and Interleukin-13 Exacerbate Neurotoxicity of Prothrombin Kringle-2 in Cortex In Vivo via Oxidative Stress

**DOI:** 10.3390/ijms20081927

**Published:** 2019-04-19

**Authors:** Jae Yeong Jeong, Young Cheul Chung, Byung Kwan Jin

**Affiliations:** 1Department of Neuroscience, Graduate School, School of Medicine, Kyung Hee University, Seoul 02447, Korea; jyoung0229@khu.ac.kr; 2Department of Biochemistry & Molecular Biology, School of Medicine, Kyung Hee University, Seoul 02447, Korea

**Keywords:** neuroinflammation, microglia, prothrombin fragment-2, interleukin-4, interleukin-13, oxidative stress

## Abstract

The present study investigated the effects of activated microglia-derived interleukin-4 (IL-4) and IL-13 on neurodegeneration in prothrombin kringle-2 (pKr-2)-treated rat cortex. pKr-2 was unilaterally injected into the Sprague–Dawley rat cerebral cortex and IL-4 and IL-13 neutralizing antibody was used to block the function of IL-4 and IL-13. Immunohistochemical analysis showed a significant loss of NeuN^+^ and Nissl^+^ cells and an increase of OX-42^+^ cells in the cortex at seven days post pKr-2. The levels of IL-4 and IL-13 expression were upregulated in the activated microglia as early as 12 hours post pKr-2 and sustained up to seven days post pKr-2. Neutralization by IL-4 or IL-13 antibodies (NA) significantly increased neuronal survival in pKr-2-treated rat cortex in vivo by suppressing microglial activation and the production of reactive oxygen species, as analyzed by immunohisotochemistry and hydroethidine histochemistry. These results suggest that IL-4 and IL-13 that were endogenously expressed from reactive microglia may play a critical role on neuronal death by regulating oxidative stress during the neurodegenerative diseases, such as Alzheimer’s disease and dementia.

## 1. Introduction

Inflammation is one of main pathological features in the diverse diseases, such as acute injury, infection, and neurodegenerative diseases [1]. Accumulating evidence indicates that the neuroinflammation, inflammatory response in the brain is considered to be the pathogenic mechanism of neurodegenerative diseases, including Alzheimer’s disease (AD) [2,3,4,5].

In general, neuroinflammation is involved in neuronal damage and loss through many biological mechanisms, such as glial activation, production of proinflammatory molecules, elevated oxidative stress, and infiltration of peripheral immune cells, including macrophages [6,7]. Among them, microglia and infiltrated macrophage are known to play a crucial role in the initiation and maintenance of neuroinflammatory conditions through change into the activated morphology [8,9,10,11]. These activated microglia and macrophages accumulate around damaged tissue and then release inflammatory cytokines and reactive oxygen species (ROS), which may cause neurodegeneration [2,11,12]. Due to microglia and macrophages sharing very similar gene expression pattern, cell surface markers, and activation states [13], this similarity limits the understanding of their pathological role during neurodegeneration. Thus, those two cell types were combined and designated as microglia/macrophages or CD11b^+^ or Tomato Lectin^+^ (TL^+^) cells.

Prothrombin Kringle-2 (pkr-2), which is a domain of prothrombin distinct from thrombin, originates from prothrombin to thrombin, prothrombin Kringle-1 by cleavage of the prothrombinase complex (Factor Xa) [14], in turn, inducing blood coagulation [15]. Increased pKr-2 and thrombin expression have been observed in the brain of patients with neurodegenerative disease, such as AD and Parkinson’s disease (PD) [16,17,18], indicating the pathological relevance of pKr-2 during neurodegeneration. Indeed, pKr-2 is able to induce the cortical [19] and dopaminergic [8] neuronal death in vitro and in vivo through microglial activation-derived neuroinflammatory responses. Toll-like receptor (TLR4), a well-known innate immune receptor in microglia, mediates this pKr-2-induced microglial activation and neurotoxicity [17]. Moreover, pKr-2 is able to induce microglial death through NADPH oxidase-derived ROS production and oxidative stress in vitro [20]. In combination with extensive findings, these results suggest that pKr-2-induced microglial activation, and neuroinflammatory responses are closely associated with the diverse pathogenesis of neurodegenerative diseases, including AD.

Interleukin-4 (IL-4) and Interleukin-13 (IL-13) are well-known anti-inflammatory cytokines that contribute to cell repair and regeneration under inflammatory condition. They interact with cells through a common receptor, consisting of IL-4Rα and IL-13Rα1, and regulate the polarization of periphery macrophage phenotype by activating STAT6 [21,22]. In the central nervous system (CNS), IL-4 and IL-13 exerts neuroprotective effects on neurons in the animal models of experimental autoimmune encephalomyelitis (EAE) [23] and AD [24] by downregulating the production of inflammatory mediators. By contrast, IL-4 and/or IL-13 potentiate LPS- and beta-amyloid-induced inflammation by producing ROS and proinflammatory cytokines, such as tumor necrosis factor alpha (TNF-α), IL-6, and inducible nitric oxide synthase (iNOS) [9,25,26,27,28]. Although detailed mechanisms for discrepant actions of IL-4 and IL-13 under neuroinflammatory conditions remain unanswered, these results suggest that IL-4 and IL-13 play a pivotal role on neuroinflammation under neuropathological environments. The current study sought to determine whether endogenous IL-4 and IL-13 expression within reactive microglia/macrophages is associated with the death of cortical neurons by regulating oxidative stress in the pKr-2 injected cortex in vivo.

## 2. Results

### 2.1. pKr-2 Induces Microglia/Macrophages Activation, Expression of Pro-Inflammatory Molecules, and Degeneration of Rat Cortical Neurons In Vivo

As pKr-2-induced microglial activation and the expression of proinflammatory molecules produce neurotoxicity on rat mesencephalic dopamine neurons in vivo [8], we wondered whether pKr-2 could cause microglia/macrophages activation, the expression of proinflammatory molecules, and the death of rat cortical neurons in vivo. We unilaterally injected pKr-2 (48 μg/4 μL) or PBS as a control into rat cerebral cortex to test this (Figure 1A). At the indicated time points, the brains were removed, and sections were processed for immunostaining using OX-42 antibody against CD11b (Figure 1A). In the PBS-injected cortex (12 h post-injection), most CD11b-immunopositive (CD11b^+^) microglia/macrophages exhibited a resting morphology of small cell bodies and long ramified processes (Figure 1A). In contrast, at 12 h after intracortical injection of pKr-2, the morphology of the CD11b^+^ microglia/macrophage exhibited the activated form with larger cell bodies and short, thick, or no processes, and this morphological change sustained for up to three days after pKr-2 injection (Figure 1A).

Afterwards, we determined whether activated microglia/macrophages by pKr-2 could be associated with the upregulation of proinflammatory cytokines (TNF-α and IL-6) and iNOS expression. RT-PCR analysis illustrated increases in expression of TNF-α, IL-6 and iNOS mRNA in the cortex in vivo as early as 12 h after pKr-2 injection (Figure 1B,C) and these increased levels were maintained for up to 72 h after pKr-2 injection. To further confirm the cellular location of these cytokines, double-immunofluorescence staining with a combination of OX-42 and iNOS, Iba-1 and TNF-α antibodies was performed (Figure 1D). Simultaneous imaging of immunofluorescence on the same tissue sections revealed that pKr-2-induced expression of TNF-α and iNOS was localized to the activated microglia/macrophages. By contrast, PBS had no effects on cytokine production (Figure 1D).

We examined whether microglia/macrophages activation and the proinflammatory molecules produced by pKr-2 could be associated with neurodegeneration. At three days after pKr-2 injection, the significant loss of neurons was detectable in the cerebral cortex, as visualized by NeuN immunostaining, as compared to PBS-treated control (Figure 1E). Nissl^+^ staining confirmed the substantial loss of cortical neurons in vivo (Figure 1E), when compared to PBS-treated control (Figure 1E). These results carefully suggest that pKr-2 induced microglia/macrophages activation and the expression of proinflammatory molecules are related to cortical neuronal loss in vivo.

### 2.2. Levels of IL-4 and IL-13 Are Increased on TL^+^ Microglia/Macrophages in pKr-2-Injected Cerebral Cortex In Vivo

Afterwards, we investigated whether pKr-2 might induce expression of IL-4 and IL-13 protein in the cerebral cortex. Immunohistochemical analysis demonstrated that pKr-2-induced expression of IL-4 (Figure 2A,B) and IL-13 (Figure 2A,C) were detected as early as one day post pKr-2, gradually increased at one day post pKr-2, and significantly increased up to seven days post pKr-2, as compared to PBS control (Figure 2A–C). To identify the cell types for IL-4 and IL-13 expressing cell in the cerebral cortex, double immunofluorescence staining with a combination of IL-4 or IL-13, and tomato lectin (TL) for microglia/macrophages, NeuN for neurons, and glial fibrillary acidic protein (GFAP) for astrocytes was performed one day after pKr-2 injection. The fluorescence images from each channel of the double-labeled sections were merged. The results showed that pKr-2-induced expression of IL-4 or IL-13 was mainly localized in TL^+^ microglia/macrophages (Figure 3A,D), but neither in NeuN^+^ neurons (Figure 3B,E) nor GFAP^+^ astrocytes (Figure 3C,F).

### 2.3. IL-4 and IL-13 Mediate Loss of Cortical Neurons in pKr-2-Tretaed Cortex In Vivo

As IL-4 and IL-13 contribute to neurodegeneration in Aβ_1-42_- [9] and thrombin- [25,26] treated rat hippocampus in vivo, we wondered whether these molecules could exert neurotoxicity on cortical neurons in pKr-2-treated cerebral cortex. Accordingly, we tested the effect of IL-4 and IL-13 on cortical neurons in vivo. IL-4 and IL-13 neutralizing antibody (IL-4NA and IL-13NA) was unilaterally co-injected with pKr-2 to block the actions of these two molecules in the ipsilateral rat cortex. The loss of NeuN^+^ (Figure 4D,E) and Nissl^+^ (Figure 4F) cells in the cortex was significantly increased in the pKr-2-lesioned rats, respectively, as compared PBS control (Figure 4A–C). This increase in neurodegeneration was significantly diminished in the pKr-2-lesioned rats that were treated with IL-4NA (Figure 4G–I) and IL-13NA (Figure 4J–L), respectively.

### 2.4. IL-4 and IL-13 Mediate Microglial Activation and ROS Production in pKr-2-Lesioned Cerebral Cortex In Vivo

IL-4 and IL-13 contribute to microglial activation and ROS production [9,29]. Accordingly, we examined whether pKr-2-induced microglia/macrophages activation and IL-4 or IL-13 might influence ROS production. Immunohistochemical analysis showed that changes in the intensity and morphology of OX-42^+^ microglia/macrophages were observed in pKr-2-lesioned rat brain (Figure 5D,E) when compared with PBS control (Figure 5A,B). These pKr-2-induced changes were reduced in the cerebral cortex that was treated with IL-4NA (Figure 5G,H) or IL-13NA (Figure 5J,K), respectively.

Activated microglia are the major source for ROS generation, which may lead to neuronal death through oxidative stress in the animal models of neurodegenerative diseases, including Alzheimer’s disease and Parkinson’s disease [6,9,25,26,29,30]. As functional inhibition of IL-4 and IL-13 seems to exert neuroprotection through inhibiting the activation of microglia/macrophages, we hypothesized that IL-4NA and IL-13NA could lessen tactivated microglia/macrophages-derived ROS production. To test this, hydroethidine histochemistry was performed for the in situ visualization of O2^−^ production (Figure 5C,F,I,L). Intracortical injection of pKr-2 increased O2^−^ production in the cerebral cortex in vivo (visualized as the fluorescent product of oxidized hydroethidine, i.e., ethidium accumulation) (Figure 5F) as compared to PBS (Figure 5C). In the pKr-2 and IL-4NA (Figure 5I) or IL-13NA (Figure 5L) treated cortex, O2^−^ production was significantly reduced when compared to pKr-2 only treated cortex at three days post injection (Figure 5F).

Double immunofluorescence staining revealed co-localization of ROS within OX-42^+^ microglia/macrophages (Figure 5M). At three days post pKr-2, treatment with IL-4NA and IL-13NA significantly attenuated the pKr-2-induced ROS production, as compared to pKr-2 only treated cortex (Figure 5N).

## 3. Discussion

The present study shows that the increased expression of IL-4 and IL-13 within activated microglia/macrophages mediates pKr-2-induced neurotoxicity in the cerebral cortex in vivo. Intracortical injection of pKr-2 results in cortical neuronal loss through activation of microglia/macrophages and production of inflammatory cytokines such as TNF-α, IL-6, and iNOS. Treatment with IL-4 and IL-13 neutralizing antibodies rescued cortical neurons against pKr-2-induced neurotoxicity and reduced microglia/macrophages activation-derived ROS production in vivo, indicating the involvement of IL-4 and IL-13. This study is first to demonstrate that endogenous IL-4 and IL-13 that were derived from microglia/macrophages contribute to pKr-2 induced-cortical neurodegeneration by regulating microglia/macrophages activation and ROS production in vivo (Figure 6).

In the coagulation cascade, pKr-2 is generated from the precursor prothrombin by cleaving to fragment 1-2 (kringle regions) and thrombin [31,32]. Prothrombin is endogenously expressed in the human, rat, and mouse brain [17,33,34], and it circulates in blood at micromolar levels [35]. Under pathological conditions, such as AD and PD, cerebrovascular damages propagate the pKr-2 production through conversion from prothrombin to thrombin, which possibly results in the extravasation of pKr-2 into the brain [36]. Recently, we reported that stereotaxic injection of pKr-2 in the cerebral cortex and substantia nigra was toxic to cortical and dopaminergic neurons in vivo [8,19]. Moreover, experimental and clinical evidence have suggested that the AD pathology is associated with abnormal blood coagulation system related with pKr-2 generation, such as an elevated level of thrombin [37] and fibrinogen deposition in the cerebral cortical parenchyma [38]. Regarding this, our unpublished observations indicate the up-regulated expression of pKr-2 in the brain (cortex and hippocampus) of 5 X Familiar AD (FAD) mouse and AD patients (data not shown). Collectively, these results suggest that pKr-2 originated from brain or blood or both is attributed to cortical neuronal loss in AD pathophysiology.

Microglia are brain-resident immune cells that closely resemble peripheral macrophage. Under pathological conditions, microglia proliferate and shift into different functional states (microglial activation) to increase the phagocytic activity, antigen presentation, and the release of inflammatory mediators and ROS [39]. Microglial activation has been implicated with the neurodegeneration and altered inflammatory response in neurodegenerative diseases, such as AD and PD. Indeed, we have shown that microglial activators, such as lipopolysaccharide (LPS) [10,40], beta-amyloid [9], and thrombin [25,26] induce the production of diverse inflammatory cytokines and ROS to control brain inflammation and/or neuronal loss in vivo. Our recent studies have demonstrated that pKr-2, another microglial activator, produces the death of dopamine neurons and cortical neurons in vivo and in vitro by the up-regulation of proinflammatory cytokines, such as iNOS, nitric oxide, and NADPH oxidase-derived ROS production [8,17,19]. This is consistent with the present results, which showed that pKr-2-induced activation of microglia/macrophages and ROS production may be involved in cortical neuronal death in vivo.

Anti-inflammatory functions of IL-4 and IL-13 are well documented. IL-4 and IL-13 suppress the production of ROS [41] and inflammatory molecules, such as IL-6, IL-1β, TNF-α, and iNOS [42]. The overexpression of IL-4 using lentivirus or mesenchymal stem cell inhibited the neuroinflammation by enhancing the anti-inflammatory response in vivo [43,44]. In an animal model of EAE, human recombinant IL-13 and regulatory T cell-derived IL-13 exert neuroprotective effect by inactivating macrophage and reducing oxidative stress [41,45]. The neutralization of IL-4 and IL-13 aggravated neuronal death by increasing activated microglia/macrophages-derived proinflammatory response in vivo in the LPS-injected cortex [10] and traumatic spinal cord injury [46]. Collectively, all of these results suggest that the IL-4 and IL-13 have anti-inflammatory properties and a regulatory effect on microglia/macrophages activation in the neuroinflammation in vivo.

On the contrary, emerging evidence has shown the neurotoxic effects of IL-4 and IL-13 under neuroinflammatory conditions. Enhanced T-cell-mediated IL-4 production promotes the atypical EAE development in vivo [47]. The genetic ablation of IL-13 prevents EAE progression by reducing the infiltration of CD11b^+^ and MHC-II^+^ cells in vivo [48]. Although exogenous IL-4 and IL-13 does not have direct toxicity on neuron, they increase the susceptibility of neurons to oxidative damage in vitro [49], indicating the harmful actions of these two cytokines. We reported that IL-4NA and IL-13NA inhibited the activation of NADPH oxidase, and/or the production of iNOS, ROS, and RNS derived from reactive microglia/macrophages in the thrombin- or Aβ1-42-treated CA1 layer of hippocampus in vivo, leading to neuronal survival [9,25,26]. In cultured rat microglia that were exposed to pKr-2 [20] or LPS [28], IL-4 and IL-13 exacerbate oxidative stress by regulating NADPH oxidase and COX-2, resulting in the cell death of microglia. All of these results are in line with our current data that IL-4 and IL-13 contribute to the degeneration of pKr-2-treated cortical neurons in vivo by enhancing microglia/macrophages activation and ROS production.

It seems noteworthy that the current results are not consistent with our previous report, which showed that endogenous IL-4 and IL-13 inhibit the microglia/macrophages activation, resulting in neuronal survival in LPS-injected cortex and spinal cord injury [10,46,50]. By contrast, in the present study, we described that both IL-4 and IL-13 participate in pKr-2-induced reactive microglia/macrophages-derived ROS production and neurodegeneration. This apparent discrepancy may be attributed to the initial neuroinflammatory environments through the use of different stimuli (LPS vs pKr-2). In the previous reports, Ji et al. demonstrated that cortical LPS injection leads to ramified CD11b^+^ resident microglial death within 6 h and most of the round CD11b^+^ cells that were detected at 12 h were blood-born neutrophils [51]. However, we did not observe the death of CD11b^+^ microglia/macrophage and neutrophil infiltration within 24 h under our experimental conditions (data not shown). Although the neuroinflammatory environments in response to pKr-2 remain to be determined, these initial differences result in the discrepant effect of IL-4 and IL-13 on cortical neurons in vivo.

To our knowledge, this is the first study to show the toxicity of IL-4 and/or IL-13 being endogenously expressed from activated microglia/macrophages after pKr-2 injection on the degeneration of cortical neurons in vivo. The current findings, in combination with many experimental results, suggest that the deleterious actions of microglia/macrophages-derived endogenous IL-4 and/or IL-13 are possibly involved in oxidative stress-mediated neurodegenerative disease, such as dementia and AD.

## 4. Materials and Methods

### 4.1. Chemicals

The materials were purchased from the following companies: pKr-2 (Haematologic Technologies Inc., Essex Junction, VT, USA), IL-4 and IL-13 neutralizing antibody (R&D systems, Minneapolis, MN, USA), and dihydroethidium (Life Technologies Corp., Eugene, OR, USA).

### 4.2. Animals

All of the experiments were done in accordance with Institutional Animal Care and Use Committee of Kyung Hee University and to minimize the number of animal experiments and suffering. We carried out the experiment with the protocols and guidelines that were established by Kyung Hee University (KHUASP (SE)-16-059, 8 August 2016). Female Sprague-Dawley rats (10 weeks of age, 240–270 g, purchased from Daehan Biolink, introduced from Taconic Co., Albany, NY, USA) were housed under a 12:12 h (hour) light: dark cycle at an ambient temperature of 22 °C. Water and rat chow were available ad libitum.

### 4.3. Stereotaxic Injection of pKr-2 and IL-4 and IL-13 Neutralization

Stereotaxic surgery under chloral hydrate (360 mg/kg, intraperitoneally) was performed, as previously described [10,40]. Using coordinates relative to the bregma, stereotaxic injections of pKr-2 (right cortex; A/P +1.4 mm, M/L −2.0mm, D/V −2.0mm; 48 μg in 4 μL phosphate-buffered saline; Gibco, Paisley, UK), or IL-4 or IL-13 neutralizing antibody (1 μg/uL), and respective control were done according to the atlas of Paxinos and Watson [10,40].

### 4.4. Immunohistochemstry and Immunofluorescence Staining

As previously described [52], the rat brain tissues were prepared for immunohistochemistry and immunofluorescence staining. The animals were transcardially perfused and fixed, and the brains were dissected out, frozen, and cut into 40-μm-thick coronal sections using a microtome (Microm HM 450, Thermo Fisher Scientific, Walldorf, Germany). Briefly, brain tissues were rinsed in PBS and then quenched in PBS containing 3% H_2_O_2_. Tissues were then rinsed in PBS and blocked for 30 min in PBS containing 1% bovine serum albumin (BSA; Rocky Mountain Biologicals Inc, Missoula, MT, USA), 0.2% triton X-100 (Sigma, St. Louis, MO, USA). Subsequently, tissues were incubated with PBS containing 0.5% BSA and each following primary antibodies: mouse anti-OX-42 (1:400, Bio-rad, Hercules, CA, USA), rabbit anti-ionized calcium binding adaptor molecule 1 (Iba-1; 1:1000, Wako, Richmond, VA, USA), and Fluorescein isothiocyanate (FITC)-labeled tomato lectin (1:1000, Vector Laboratories, Burlingame, CA, USA) for microglia/macrophages, mouse anti-neuron-specific nuclear protein (NeuN; 1:400, Merck Millipore, Darmstadt, Germany) for general neurons, mouse-anti-glial fibrillary acidic protein (GFAP; 1:500, Sigma) for astrocytes, mouse rabbit-anti-inducible nitric oxide (iNOS; 1:200, BD Biosciences, San Jose, CA, USA), mouse anti-tumor necrosis factor-α (TNF-α; 1:200, R&D Systems, Minneapolis, MN, USA), goat-anti-Interleukin-4 (IL-4; 1:400, R&D Systems), and goat anti-IL-13 (1:200, R&D Systems). After incubation, the brain tissues were incubated with biotinylated secondary antibodies, biotin-conjugated mouse (1:400, KPL, Milford, MA, USA), or rabbit (1:400, Vector Laboratories), followed by the addition of the avidin-biotin reagent (1:1:100, Vector Laboratories) or incubated with fluorescence-conjugated secondary antibodies (FITC-conjugated anti-mouse (1:500, Merck Millipore), Cy3-conjugated anti-rabbit (1:1000, Merck Millipore), and Alexa Fluor 594-conjugated anti-goat IgG (1:1000, Molecular Probes, Eugene, OR, USA). The signal following treatment with avidin-biotin reagent was detected by incubating the sections in 0.5 mg/mL 3,3’-diaminobenzidine (DAB; Sigma) in 0.1 M PB containing 0.003% H_2_O_2_. The tissues were viewed under a bright-field microscope (Olympus Optical, Tokyo, Japan). The tissues incubated with fluorescence-conjugated antibodies were washed, covered with Vectashield medium (Vector Laboratories), and analyzed under confocal microscopy (LSM700, Carl Zeiss, Oberkochen, Germany). For Nissl staining, the tissues were mounted on coated slides, dried for 1hr, stained with 0.5% cresyl violet acetate (Sigma), dehydrated, covered, and then viewed under a bright-field microscope.

### 4.5. In Situ Detection of O_2_^−^ and O_2_^−^-Derived Oxidants

Hydroethidine histochemistry was performed for in situ visualization of the O_2_- and O_2_-derived oxidants. Three days after pKr-2 injection, hydroethidine (1 mg/kg in PBS containing 1% dimethylsulfoxide; Sigma, St. Louis, MO, USA) was intravenously administered through tail vein. After 45 min from hydroethidine injection, the brain tissues were prepared, as previously described [40,52]. The brain tissues were cut into 40 µm using a sliding microtome (Thermo Scientific, Walldorf, Baden-Württemberg, Germany) and the tissues were mounted on gelatin-coated slides. The oxidized hydroethidine product, ethidium, examined by confocal microscopy (Carl Zeiss), and then merged with DAPI solution (Vector Laboratories). Image J quantified the obtained images in each group (National Institutes of Health, USA).

### 4.6. Reverse-Transcription Polymerase Chain Reaction (RT-PCR) for Cytokines

As previously described [10,40], brain tissues from the ipsilateral cortex were dissected and immediately isolated at each time point after pKr-2 injection. Total RNA was prepared with RNAzol B (Tel-Test, Friendwood, TX, USA) and reverse transcription was carried out using Superscript II reverse transcriptase (Life Technologies Corp.). The primers used for TNF-α, IL-6, and iNOS were as follows: 5’-GTA GCC CAC GTC GTA GCA AA-3’ (sense) and 5’-CCC TTC TCC AGC TGG GAG AC-3’ (antisense) for TNF-α, 5’-AAA ATC TGC TCT GGT CTT CTG G-3’ (sense), and 5’-GGT TTG CCG AGT AGA CCT CA-3’ (antisense) for IL-6, 5’-GCA GAA TGT GAC CAT CAT GG-3’ (sense), and 5’-ACA ACC TTG GTG TTG AAG GC-3’ (antisense) for iNOS. The PCR cycles consisted of denaturation at 94 °C for 30 s, annealing at 55 °C for 30 s (TNF-α, iNOS) or 60 °C for 30 s (IL-6), and extension at 72°C for 90 s for 30 cycles. The PCR product was separated by electrophoresis on a 1.5% agarose gel, stained with ethidium bromide, and then detected under UV light.

### 4.7. Image J Analysis

To analyze and quantify the expression of IL-4, IL-13, and ethidium in the pKr-2-injected cortex, images were obtained from the same area in each tissue samples. Imaging data were analyzed in Image J (National Institutes of Health, Bethesda, MD, USA). Image J was used to quantify the chromogenic signal intensity of immunofluorescence on image.

### 4.8. Statistical Analysis

All of the values are expressed as mean standard error of the mean. Statistical significance (*p* < 0.05 for all analysis) was assessed by One way ANOVA Newman–Keuls analyses and Student unpaired *t*-test (GraphPad Software, San Diego, CA, USA).

## Figures and Tables

**Figure 1 ijms-20-01927-f001:**
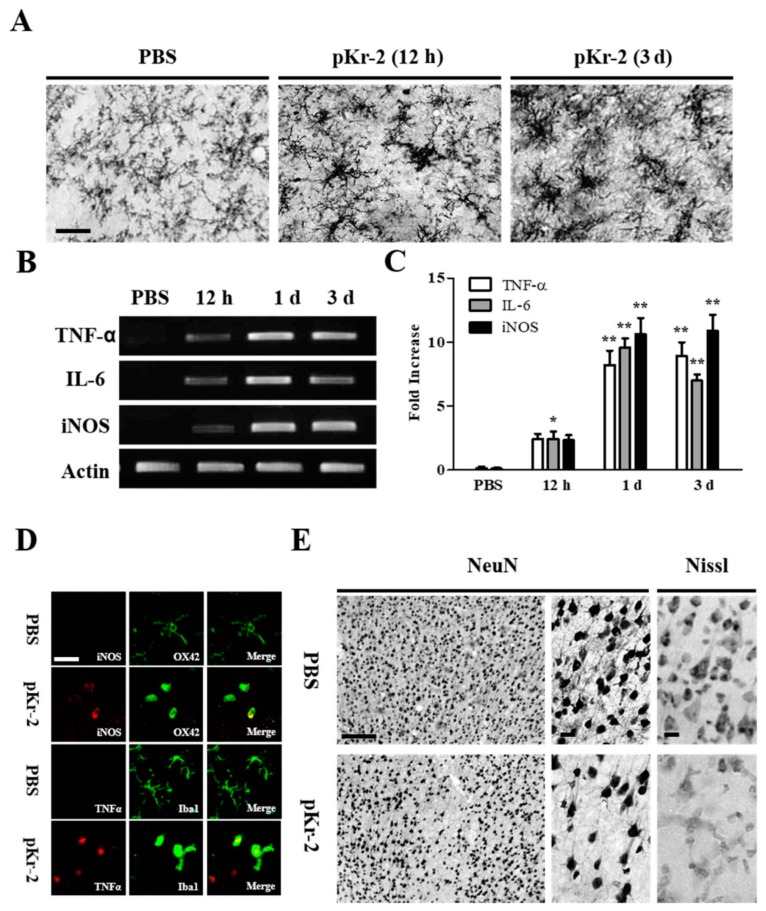
Microglial activation by prothrombin kringle-2 (pKr-2) leads to death of cortical neurons by increasing expression of proinflammatory cytokines in vivo. (**A**–**C**) pKr-2 (48 μg/4 μL) or PBS as a control was unilaterally injected into the cerebral cortex. Animals were killed at indicated time point, brains tissues were prepared for immunohistochemical staining and Reverse-Transcription Polymerase Chain Reaction (RT-PCR) analysis. For the immunohistochemistry, coronal sections (40 μm) were cut using sliding microtome. Every sixth serial sections were selected and processed for OX-42 immunostaining for microglia. (**A**) Note morphological changes in OX-42^+^ microglia from resting (small cell bodies and thin, long, or ramified processes) to activated state (larger cell bodies with short processes). (**B**,**C**) RT-PCR analysis of anti-tumor necrosis factor-α (TNF-α), Interleukin-6 (IL-6), and inducible nitric oxide synthase (iNOS) (**B**), and quantification (**C**) in the cerebral cortex of pKr-2-injected rat brain at the indicated time points. Error bars represent the mean ± SEM. * *p* < 0.05, ** *p* < 0.001 compared with control according to Student *t*-Test and One way ANOVA and Newman-Keuls analyses. (**D**) Fluorescence images showing co-localization of iNOS (red) and OX-42^+^ microglia (green) or TNF-α (red) and Iba1^+^ microglia (green) in the cerebral cortex at one day after intracortical injection of pKr-2 or PBS as a control. (**E**) Immunohistochemical staining showing loss of NeuN^+^ and Nissl^+^ in the cerebral cortex at three days after intracortical injection of pKr-2 or PBS as a control. Scale bars, **A**, 150 μm. **D**, 20 μm, **E**, 25 μm; **A**, *n* = 4 to 6; **B**,**C**, *n* = 4; **D**, *n* = 4; **E**, *n* = 4 to 6.

**Figure 2 ijms-20-01927-f002:**
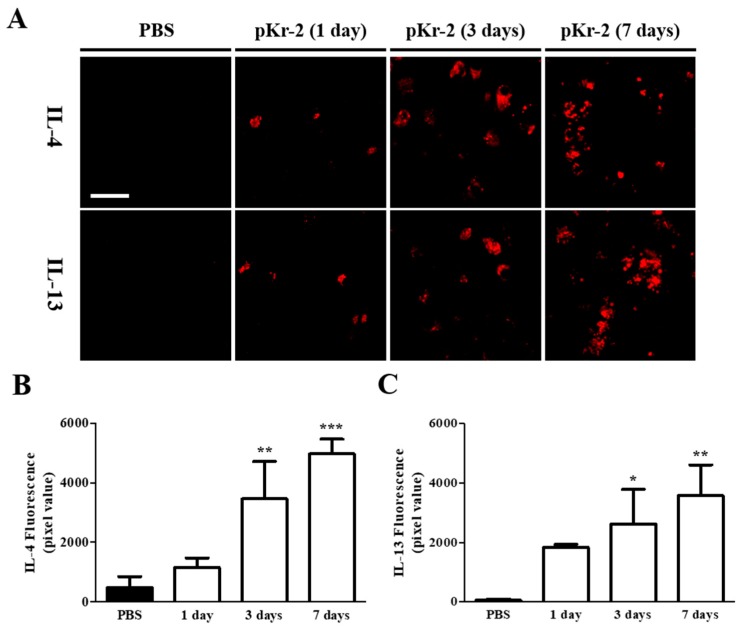
Intracortical injection of pKr-2 induces an increase of Interleukin-4 (IL-4) and Interleukin-13 (IL-13) expression in vivo. Cerebral cortical tissue sections, adjacent to those used in Figure 1 were immunostained with IL-4 and IL-13. (**A**) Fluorescence images of IL-4 and IL-13, and (**B**,**C**) quantification in the cerebral cortex using Image J at the indicated time points. Error bars represent the mean ± SEM. * *p* < 0.05, ** *p* < 0.01, *** *p* < 0.001 as compared with control according to One way ANOVA and Newman–Keuls analyses. Scale bar, 40 μm; *n* = 3 to 6.

**Figure 3 ijms-20-01927-f003:**
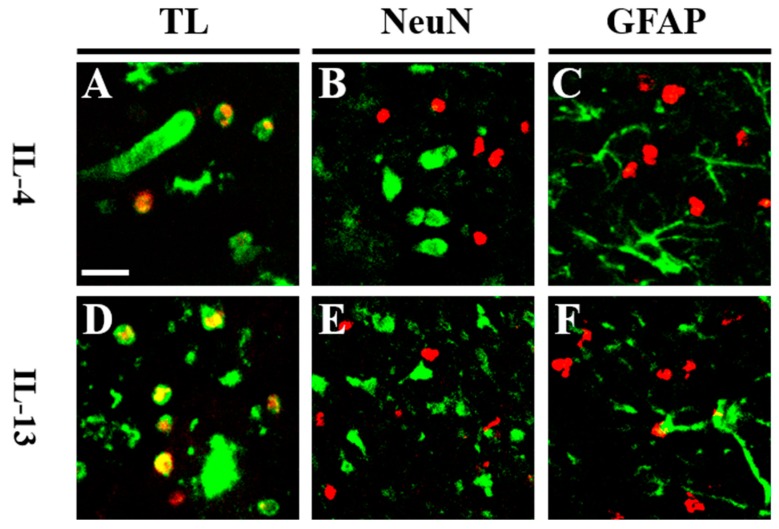
pKr-2-induced IL-4 and IL-13 are co-localized within activated microglia/macrophages in vivo. (**A**–**F**) Animals receiving a unilateral injection of pKr-2 into cerebral cortex were sacrificed 1 day later, brains were removed, and coronal sections (40 μm) were prepared for immunohistochemical staining. Fluorescence images of (**A**,**D**) Tomato Lectin (green) for microglia/macrophages and IL-4 (**A**, red) or IL-13 (**D**, red), (**B**,**E**) NeuN (green) for neurons and IL-4 (**B**, red) or IL-13 (**E**, red), and (**C**,**F**) glial fibrillary acidic protein (GFAP) (green) for astrocytes and IL-4 (**C**, red), or IL-13 (**F**, red). Each image was captured from the similar cortical area and merged (yellow). Scale bar: 25 μm. *n* = 4 to 6. TL: tomato lectin.

**Figure 4 ijms-20-01927-f004:**
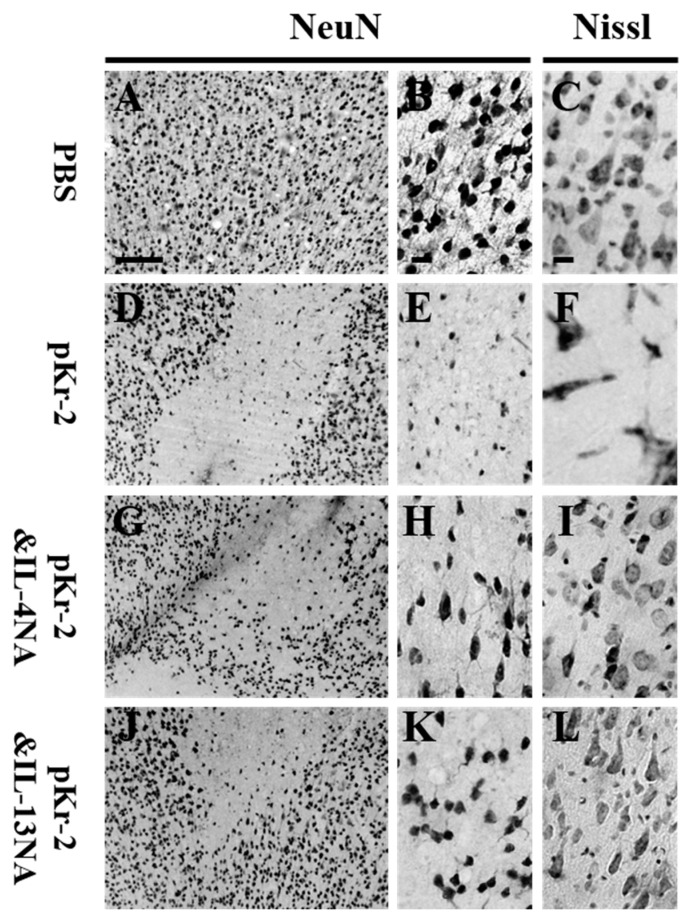
Neutralization of IL-4 or IL-13 prevents degeneration of cortical neurons in vivo in the pKr-2-injected rat brain. (**A**–**L**) PBS (4 μL; **A**–**C**) or pKr-2 (48 μg/4 μL; **D**–**F**) was unilaterally injected into cerebral cortex in the presence of IL-4NA (1 µg; **G**–**I**) or IL-13NA (1 µg; **J**–**L**). At 7 d post-injection, the coronal sections were selected and processed for NeuN (**A**,**B**,**D**,**E**,**G**,**H**,**J**,**K**) immunohistochemical staining or Nissl staining (**C**,**F**,**I**,**L**). (**B**,**E**,**H**,**K**) Higher magnification of **A**,**D**,**G** and **J**, respectively. Scale bars: 250 μm (**A**) and 20 μm (**B**,**C**). *n* = 4 to 6.

**Figure 5 ijms-20-01927-f005:**
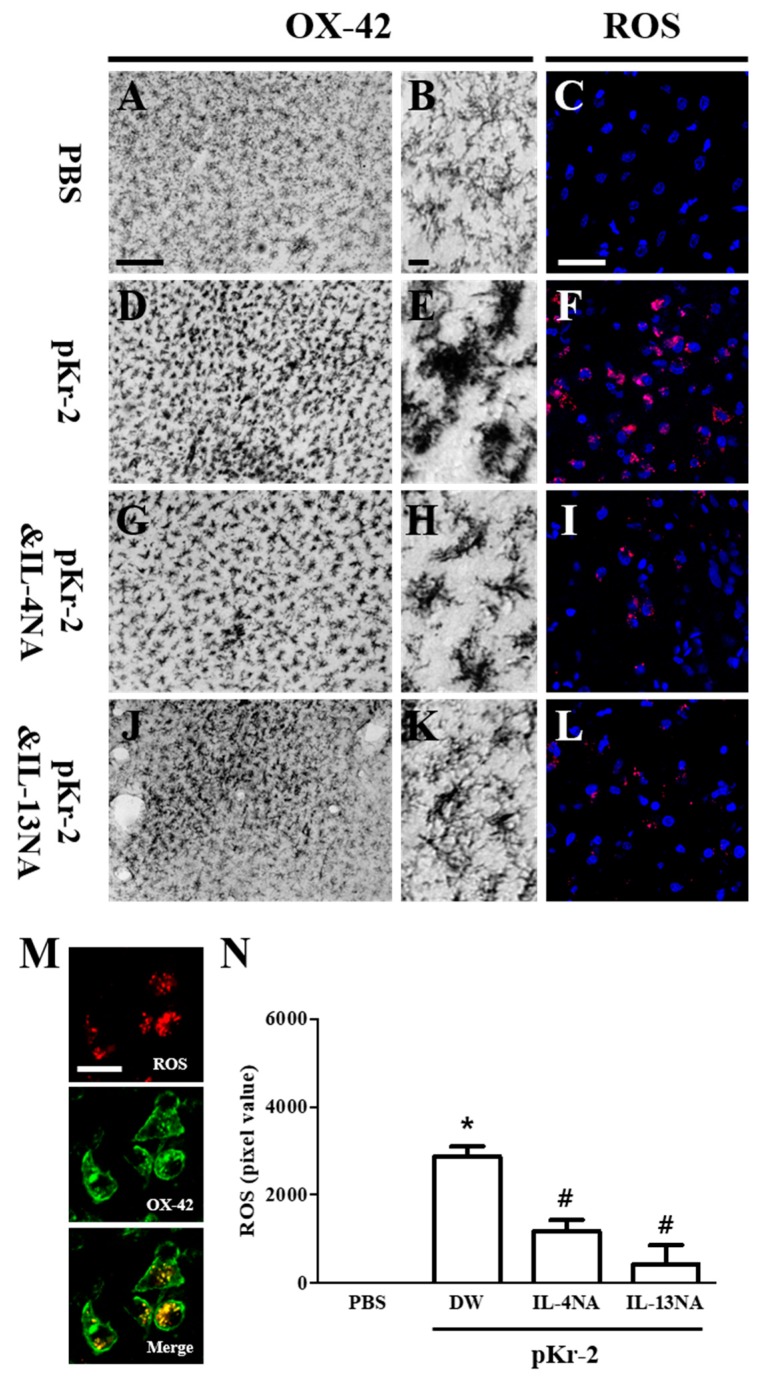
Neutralization of IL-4 or IL-13 inhibits the microglia/macrophages activation and reactive oxygen species (ROS) production in vivo. (**A**–**L**) Animals receiving PBS as a control (**A**–**C**), pKr-2 alone (**D**–**F**), pKr-2 and IL-4NA (**G**–**I**), and pKr-2 and IL-13NA (**J**–**L**) were sacrificed three days after the intracortical injection. Brain tissues were cut and cortical tissues were immunostained with an antibody for OX-42 to label microglia/macrophages (**A**,**B**,**D**,**E**,**G**,**H**,**J**,**K**) and were prepared for hydroethidine histochemistry to detect oxidant production (**C**,**F**,**I**,**L**). (**A**,**B**,**D**,**E**,**G**,**H**,**J**,**K**) Photomicrograph of OX-42^+^ cells in cerebral cortex in vivo. (**B**,**E**,**H**,**K**) Higher magnification of **A**, **D**, **G** and **J**, respectively. (**C**,**F**,**I**,**L**) Confocal microscope show ethidium fluorescence (red). Nuclei were counterstained with DAPI (blue). (**M**) Fluorescence images showing co-localization of ROS (red) and OX-42^+^ microglia/macrophages (green) in the cerebral cortex at 3 d after intracortical injection of pKr-2. (**N**) Quantification of ROS expression. Error bars represent the mean ± SEM. * *p* < 0.001, as compared with control, # *p* < 0.001, compared with pKr-2 & DW according to One way ANOVA and Newman–Keuls analyses. Scale bars: 150 μm (**A**), 20 μm (**B**), 50 μm (**C**), 20 μm (**M**); *n* = 5 to 7 (**A**,**B**,**D**,**E**,**G**,**H**,**J**,**K**), *n* = 3 to 5 (**C**,**F**,**I**,**L**,**N**).

**Figure 6 ijms-20-01927-f006:**
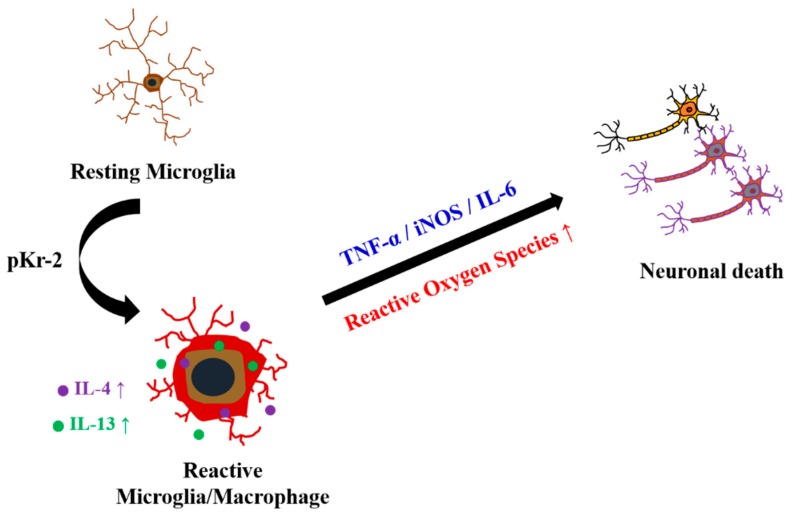
Interleukin-4 (IL-4) and IL-13 induced oxidative stress contributes to death of cortical neurons in pKr-2-treated cerebral cortex in vivo. pKr-2, Prothrombin kringle-2; IL-4, Interleukin-4; IL-13, Interleukin-13; TNF-α, Tumor necrosis factor-α; iNOS, Inducible nitric oxide synthase; IL-6, Interleukin-6.

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
