# Peer review of "Interleukin-4 and Interleukin-13 Exacerbate Neurotoxicity of Prothrombin Kringle-2 in Cortex In Vivo via Oxidative Stress"

_ijms, 2019, doi:10.3390/ijms20081927_

Round 1
Reviewer 1 Report
Jaeyeong Jeong et al., in this current manuscript showed the neurotoxic effects of Prothrombin Kringle-2 in cortex of rat under in vivo settings. Authors showed that these toxic effects were dependent on the microglia driven interleukin 4 and interleukin 13. To my opinion experiments are well designed and well performed. Thus, I encouraged this manuscript to be accepted with few modification as mentioned below.
1. In fig. 1C authors did not provide information by using signs of significance on 12 hr point if these were significantly different as shown in other time points.
2. On page 2, Line 44 author wrote TL+ cells, pls expand this abbreviation here already as reader might not know what you meant here until they reach to page number 4.
3. Authors are encouraged to draw cartoon (showing the cell types and moleciules involved) to explain the mechanism by which PKr2 exerts its toxic effects on neurons and how IL-4 and IL-13 neutralizing antibodies protects this action.
Author Response
Reviewer 1.
Jaeyeong Jeong et al., in this current manuscript showed the neurotoxic effects of Prothrombin Kringle-2 in cortex of rat under in vivo settings. Authors showed that these toxic effects were dependent on the microglia driven interleukin 4 and interleukin 13. To my opinion experiments are well designed and well performed. Thus, I encouraged this manuscript to be accepted with few modification as mentioned below.
Comment 1. In fig. 1C authors did not provide information by using signs of significance on 12 hr point if these were significantly different as shown in other time points.
Correction 1: The statistical significance was analyzed in detail. In Student t-Test, IL-6 expression is significantly different from control at 12 hr time point. We added the information of statistical significance in Figure 1C and Figure legends 1.
Page 4, Line 111-112: In the Figure legends section, we added “*P<0.05, **P< 0.001 compared with control according to Student t-Test and One way ANOVA and Newman-Keuls analyses.”
Comment 2. On page 2, Line 44 author wrote TL+ cells, pls expand this abbreviation here already as reader might not know what you meant here until they reach to page number 4.
Correction 2: In page 2, line 44, we added “Tomato Lectin+ (TL+)“ cells.
3. Authors are encouraged to draw cartoon (showing the cell types and molecules involved) to explain the mechanism by which PKr2 exerts its toxic effects on neurons and how IL-4 and IL-13 neutralizing antibodies protects this action.
Correction 3: A new cartoon was added as Fig. 6 in order to show how pKr-2 exerts its toxic effects on neurons and how IL-4 and IL-13 exacerbate microglia/macrophages-induced neurotoxicity.
Page 9, Line 207: In Figure legends section, we added “Fig. 6. Interleukin-4 (IL-4) and IL-13 induced oxidative stress contributes to death of cortical neurons in pKr-2-treated cerebral cortex in vivo. pKr-2, Prothrombin kringle-2; IL-4, Interleukin-4; IL-13, Interleukin-13; TNF-α, Tumor necrosis factor-α; iNOS, Inducible nitric oxide synthase; IL-6, Interleukin-6.

Reviewer 2 Report
Article is interesting, however one item requires some considerations. Namely, what are authors
idea about discrepancy between ani-inflammatory functions of IL-4 and IL-13 in mentioned literature data and new findings indicating contribution of these ILs in pKr-2-induced neuronal damage.Is pKr-2 so toxic that IL-4 and IL-13 are not able to protect neuroinflammation, or other reasons?. It would be reasonable to add in CONCLUSIONS at least some sentences.
Author Response
Reviewer 2.
Article is interesting, however one item requires some considerations. Namely, what are authors idea about discrepancy between anti-inflammatory functions of IL-4 and IL-13 in mentioned literature data and new findings indicating contribution of these ILs in pKr-2-induced neuronal damage. Is pKr-2 so toxic that IL-4 and IL-13 are not able to protect neuroinflammation, or other reasons? It would be reasonable to add in CONCLUSIONS at least some sentences.
Correction 1: In the Discussion section, we discussed how IL-4 and IL-13 are toxic in the pKr-2-induced neurotoxicity.
Page 10, Line 261: In the Discussion section, we added “It seems noteworthy that the current results are not consistent with our previous report showing that endogenous IL-4 and IL-13 inhibit the microglia/macrophages activation, resulting in neuronal survival in LPS-injected cortex and spinal cord injury [10, 46, 50]. By contrast, in the present study, we described that both IL-4 and IL-13 participate in pKr-2-induced reactive microglia/macrophages-derived ROS production and neurodegeneration. This apparent discrepancy may be attributed to the initial neuroinflammatory environments by the use of different stimuli (LPS vs pKr-2). In the previous reports, Ji et al demonstrated that cortical LPS injection leads to ramified CD11b+ resident microglial death within 6 hr and most of round CD11b+ cells detected at 12 hr were blood‐born neutrophils [51]. However, we did not observe the death of CD11b+ microglia/macrophage and neutrophil infiltration within 24 hr under our experimental conditions (data not shown). Although the neuroinflammatory environments in response to pKr-2 remain to be determined, these initial difference results in discrepant effect of IL-4 and IL-13 on cortical neurons in vivo.”
